# Super-Tough and Biodegradable Poly(lactide-co-glycolide) (PLGA) Transparent Thin Films Toughened by Star-Shaped PCL-*b*-PDLA Plasticizers

**DOI:** 10.3390/polym15122617

**Published:** 2023-06-08

**Authors:** Jieun Jeong, Sangsoo Yoon, Xin Yang, Young Jun Kim

**Affiliations:** 1School of Chemical Engineering, Sungkyunkwan University, Suwon 16419, Republic of Korea; juj0007@naver.com (J.J.);; 2Key Laboratory for Light-Weight Materials, Nanjing Tech University, Nanjing 210009, China

**Keywords:** fully degradable, super-tough, PLGA, star-shaped PCL-*b*-PDLA plasticizers, stereocomplexation, interfacial adhesion

## Abstract

To obtain fully degradable and super-tough poly(lactide-co-glycolide) (PLGA) blends, biodegradable star-shaped PCL-*b*-PDLA plasticizers were synthesized using natural originated xylitol as initiator. These plasticizers were blended with PLGA to prepare transparent thin films. Effects of added star-shaped PCL-*b*-PDLA plasticizers on mechanical, morphological, and thermodynamic properties of PLGA/star-shaped PCL-*b*-PDLA blends were investigated. The stereocomplexation strong cross-linked network between PLLA segment and PDLA segment effectively enhanced interfacial adhesion between star-shaped PCL-*b*-PDLA plasticizers and PLGA matrix. With only 0.5 wt% addition of star-shaped PCL-*b*-PDLA (Mn = 5000 g/mol), elongation at break of the PLGA blend reached approximately 248%, without any considerable sacrifice over excellent mechanical strength and modulus of PLGA.

## 1. Introduction

In recent years, biodegradable materials based on poly (lactic acid, poly (lactic-co-gly colic acid) (PLGA) and poly(ε-caprolactone) (PCL) have attracted significant attention for medical applications due to their appropriate mechanical properties, biodegradation, biocompatibility, and manageability [1,2,3,4,5].

These synthetic polymers have demonstrated low toxicity and received approval from the Food and Drug Administration (FDA) [6,7,8]. Their low toxicity is the main factor driving their biomedical applications.

Among these synthetic polymers, PLGA copolymer, which consists of PLA and PGA, has especially attracted attention because its mechanical properties and degradation period could be controlled by adjusting compositions of PLA and PGA [9,10]. For that reason, PLGA has been extensively applied in various fields, including orthopedic fixation materials [11,12], guided bone regeneration (GBR) [13,14], artificial skin [15,16], wound dressing [17,18] and drug carriers [19,20]. An ideal material with high strength, stiffness, and toughness would be desirable for these applications [21,22,23]. However, the extensively application of PLGA has been hindered by its inherent brittleness and short degradation period, which cannot be applied as medical tissue materials. To improve these drawbacks of biodegradable aliphatic polyesters, many strategies have been introduced via copolymerization [24], physical blending with plasticizers [11], composites [25,26,27], or rubbery elastomers [28,29], and addition of nanofillers [15,30]. Among these methods, the most commonly conducted method is combination of soft and ductile composite materials which include synthetic/natural materials [23,25,29] and organic/inorganic fillers or plasticizers [26,31,32,33,34]. This procedure is easy and simple. It also could significantly improve properties at a low fraction (<10 wt%) [35,36,37]. Through the combination of composite, polymer can show enhanced thermal, mechanical, and electrical properties. In general, this improvement method of polymers depends on the size [36], shape [38,39], dispersion of composites [40], and interfacial adhesion on the surface of composites [41]. Among these factors, the interfacial adhesion between main matrix and plasticizer is a major factor for toughening PLGA polymer as it governs the cavitation in the main matrix [42,43]. A strong interfacial adhesion can intensify the internal cavitation of dispersed rubber phase in the matrix and reduce macromolecular entanglements which can significantly alter mechanical properties, whereas a poor interfacial adhesion is responsible for debonding interfacial between polymer matrix and dispersed rubber phase at the interface.

Poly(ε-caprolactone) (PCL) has been widely introduced into biodegradable polymers by blending PCL with polymer or filling fabricated PCL composite with other polymers to enhance to mechanical properties [33,44,45,46,47]. PCL has not only excellent biocompatibility and bioactivity, but also highly rubbery and ductile characteristics 1 [48,49]. It also has a low melting point which makes materials easily fabricated in a manufacturing process. Nevertheless, between PCL and PLA, they are thermodynamically incompatible with each other [42,47]. Blending with PCL and PLA can cause poor mechanical properties due to poor interfacial adhesion and phase separation between immiscible PCL and PLA [48]. For these reasons, a number of studies have been conducted to improve the immiscibility and interfacial adhesion via addition of polymeric compatibilizers, addition of nanofillers, and preparation of newly formed grafted copolymers by chemical reaction. Ji-Mahdi Forouharshad and co-workers have investigated to PLLA and PCL blends using high surface area graphite (HSAG) in the molten PCL [49]. Mian-miao Xie and colleagues have researched electro-active shape memory properties of poly(ε-caprolactone)/functionalized multiwalled carbon nano-tube nanocomposite [50]. From these research studies, researchers could improve mechanical properties by using reinforced composites made of inorganic materials. However, inorganic composites tend to show side effects on human body due to their degradation process.

In this study, in consideration of the degradation of PLGA materials in biomedical applications, we organized biodegradable star-shaped PCL-*b*-PDLA plasticizers using natural originated xylitol as initiator to be fully biodegradable in an environmental condition. The present study describes enhanced toughness effect of PLGA (70/30) combining with star-shaped PCL-*b*-PDLA. Toughness and ductility of biodegradable PLGA were achieved effectively by stereocomplexation strong cross-linked network between PLGA and PCL-*b*-PDLA. Mechanical performance, interfacial compatibilization, and phase morphology of PCL-*b*-PDLA composite with PLGA were investigated in this work.

## 2. Materials and Methods

### 2.1. Materials

Glycolide, L-lactide, 1,4-butanediol and xylitol as initiator, and tin (II) 2-ethylhexanoate (Sn(Oct)_2_) as catalyst were purchased from Sigma-Aldrich (Seoul, Republic of Korea). ε-caprolactone (>99.0% by GC) was purchase from TCI (Seoul, Republic of Korea). D-(+)-Lactide (>99.5% by GC) was purchased from Tianjin ExceedBio Co., Ltd. (Tianjin, China). Chloroform, 1,1,1,3,3,3-hexafluoro-2-propanol (HFIP) and methanol (>99.5%) were obtained from Daejung Chemical & Metal Co., Ltd. (Busan, Republic of Korea). Before the synthesized, ε-caprolactone, 1,4-butanediol and solvents were vacuum distilled and stored in glove box. Glycolide and lactide were recrystallized, dried under vacuum and stored at 4 °C with sealing.

### 2.2. Experimental

#### 2.2.1. Synthesis of Random Poly(lactide-co-glycolide) Copolymer (PLGA)

As shown in Figure 1, random polyglycolide-*b*-polylactide copolymer (PLGA) was synthesized by ring-opening polymerization. L-lactide, glycolide, 1,4-butanediol, and Sn (Oct)_2_ were charged in a three-neck round bottom flask equipped with mechanical stirrer under vacuum atmosphere. Before heating, the flask was flushed with dry nitrogen several times and kept in a nitrogen flow condition. The flask was then immersed and stirring in a preheated oil bath for 1 h at 120 °C. The mixture was stirred and heated at 140 °C for 6 h. After 6 h, the mixture was cooled down to room temperature and dissolved in dried chloroform. The polymer was precipitated in chilled methanol and dried in a vacuum oven for 24 h at 80 °C.

#### 2.2.2. Synthesis of Star-Shaped PCL-*b*-PDLA Plasticizers

Predetermined amount of pre-dried xylitol as initiator, ε-caprolactone and Sn (Oct)_2_ as catalysts were charged in a torch dried 3-neck round bottom flask equipped with mechanical starrier and vacuum line using a glove box. The flask was sealed under vacuum and immersed in silicone oil at 140 °C. The flask was then stirred for 24 h at 140 °C. After 24 h, the flask was cooled down to room temperature. The polymer was dissolved in chloroform, precipitated in chilled methanol, filtered, and dried under vacuum at 60 °C for 24 h. Xylitol-based PCL prepolymer was used as macro-initiator for D-lactide. Predetermined amount of the prepolymer was charged again in a 3-neck round bottom flask equipped with mechanical starrier and vacuum line using a glove box. The flask was then immersed in silicone oil at 140 °C. When prepolymer was melted, predetermined amount of D-lactide was charged along with a catalyst. The flask was vacuumed and sealed again. The reaction mixture was heated for 12 h at 140 °C. It was then cooled down to room temperature, dissolved in chloroform, precipitated in chilled methanol, filtered, and dried under vacuum at 60 °C for 24 h.

#### 2.2.3. Preparation of PLGA/Star-Shaped PCL-*b*-PDLA Thin Films

PLGA and corresponding amount of the plasticizer were dissolved in pre-dried chloroform and stirred overnight. The blended solution was poured into glass plates placed in a fume-hood. Films were kept at room temperature for two days and then vacuum dried for 24 h at 40 °C to ensure complete removal of the solvent.

#### 2.2.4. Characterization

Using gel permeation chromatography (GPC), molecular weights of synthesized PLGA and star-shaped PCL-*b*-PDLA were analyzed. Gel permeation chromatography (GPC) measurements were carried out on Waters Alliance 2695/2414D equipped with a PTEF filter 0.2 μm and injection volume of 100 microliters. Calibration was done with polystyrene reference materials (Agilent Eascical PS-1). THF (1.0 mL/min) was used as the eluent. ^1^H-NMR spectroscopic measurements were performed using a Bruker Avance III 500 MHz with a cryo-platform to determine compositions of synthesized PLGA copolymer and PCL-*b*-PDLA composite. Fourier transform infrared spectroscopy (FT-IR) spectra were recorded using a NicoletTM iSTM 50. Spectra were recorded using a spectral width ranging from 500 to 4000 cm^−1^ and an accumulation of 128 scans. Tensile strength and elongation at break of PLGA films were determined with a universal testing machine (UTM). A load of 250 N from a Lloyd/LR30K at room temperature (25 ± 2 °C) was applied. Before determination, PLGA and PLGA/star-shaped PCL-*b*-PDLA films were stored in a standard environment for 24 h and cut into 50 mm × 10 mm × 0.1 mm rectangles. The speed of measurement was set at 5 mm/min with a load of 250 N. After measurement, all collected data were averaged and stress-strain curves were plotted. Thermal properties of PLGA were characterized by differential scanning calorimetric (DSC) and thermogravimetry (TGA/DTA) measurements. Thermal transitions of PLGA films were characterized by DSC using an AutoQ20 made by TA Instrument (New Castle, DE, USA). Before measurements, all samples were pre-weighed and loaded into the sample pan of the instrument. Samples were heated at a rate of 10 °C min from 25 °C to 250 °C cooled to 25 °C and reheated to 250 °C in a dry nitrogen atmosphere. Thermal stability was characterized using a TGA of Perkin Elmer (Waltham, MA, USA). Samples were loaded into the sample pan and heated at 10 °C min to 600 °C under a nitrogen atmosphere. Morphologies of PLGA and PLGA blended with star-shaped PCL-*b*-PDLA biocomposites were characterized using a scanning electron microscope (SEM) (JSM-7600F) at a power of 5.0 KV.

## 3. Results and Discussion

### 3.1. Polymerization and Characterization of PLGA

Poly(L-lactic-co-glycolic acid) (PLGA) was synthesized via ring-opening polymerization to prepare super-tough PLGA film. The reaction is presented in Figure 1. The synthesis result of PLGA molecular weight is presented in Figure 1.

Chemical compositions of PLLA and PGA were analyzed using ^1^H-NMR spectroscopy. Following ^1^H-NMR spectroscopy, the percentages of LA unit and GA unit contained in the PLGA was calculated from the integral area of the proton signals in the ^1^H-NMR spectra. Copolymer chemical compositions were calculated from the integral area of the multiplets in the ^1^H-NMR spectra at 5.08 ppm to 5.22 ppm, assigned to -CH_2_ protons of GA, and at 5.40 ppm to 5.57 ppm assigned to -CH protons of LA units, and 1.65 ppm to 1.92 ppm assigned to -CH_3_ protons of LA units.

Theoretical targeting molar ration of LA unit and GA unit were 70% and 30%. The practical percentages of LA unit and GA unit contained in the PLGA were 85% and 15%, respectively, based on the calculation. The following equations were used to determine molar percentages of LA unit and GA unit in the copolymer using ^1^H-NMR spectra of PLGA (Figure 2). The following equations were used to determine molar percentages of LA unit and GA unit in the copolymer using ^1^H-NMR spectra of PLGA [51]:(1)$$L-\mathrm{lactide}\;\mathrm{monomer}\;\%=\left(\frac{({\mathrm{CH}}_{3})\mathrm{intergral}\;\mathrm{area}/3}{({\mathrm{CH}}_{3})\mathrm{intergral}\;\mathrm{area}/3+({\mathrm{CH}}_{2})\mathrm{intergral}\;\mathrm{area}/2}\right)\times 100$$
(2)$$\mathrm{Glycolide}\;\mathrm{monomer}\;\%=\left(\frac{({\mathrm{CH}}_{2})\mathrm{intergral}\;\mathrm{area}/2}{({\mathrm{CH}}_{3})\mathrm{intergral}\;\mathrm{area}/3+({\mathrm{CH}}_{3})\mathrm{intergral}\;\mathrm{area}/2}\right)\times 100$$

### 3.2. Preparation and Characterization of Star-Shaped Poly(ε-caprolactone-co-D-lactide)

A series of star-shaped PCL-*b*-PDLA were synthesized with molecular weights of 5000, 10,000, and 15,000 to prepare biodegradable rubbery plasticizer via ring-opening polymerization using as xylitol initiator. The reaction is presented in Figure 2. Its molecular weight results are presented in Table 1. The molecular weight and distribution of star-shaped poly(ε-caprolactone-co-D-lactide) block-copolymer were determined using GPC analysis. Molecular weights of the prepared PCL-*b*-PDLA by GPC were 5.38 × 10^3^, 1.09 × 10^4^, and 1.58 × 10^4^ with PDI values of 1.14, 1.09, and 1.14, respectively, as shown Table 1.

The chemical structure of star-shaped poly(ε-caprolactone-co-D-lactide) was confirmed by ^1^H-NMR. Figure 3 shows ^1^H-NMR spectra of star-shaped poly(ε-caprolactone-co-D-lactide) with molecular weights of 5000, 10,000, and 15,000. In Figure 3, it showed predominant PCL signal in star-shaped PCL-*b*-PDLA copolymer is peak at range from $H_{a}$ to $H_{e}$ signal. The peak at 4.10 ppm ($H_{a}$) was observed due to -CH_2_ of ε-CL and the peak at 2.34 ppm ($H_{e})$ was observed due to CH_2_ of ε-CL. Peaks corresponding to CH_2_ of ε-CL were confirmed at 1.30–1.65 ppm, indicating $H_{b}$, $H_{c},\;\mathrm{and}$ $H_{d}$, respectively [52,53].

Regarding ^1^H-NMR spectra, methine group protons of PDLA showed a quadruplet signal $H_{f}$ from 5.10 ppm to 5.25 ppm [54]. In addition, methine proton peaks belonging to hydroxy group(-CH-OH) in PDLA chain showed a peak $H_{h}$ at 4.17 ppm (Figure 3) [55,56]. The double signal at 1.4 ppm corresponding to methyl protons of PDLA main chains also appeared, which further proved that PDLA block was connected with PCL [56]. Thus, the formation of star-shaped poly(ε-caprolactone-co-D-lactide) was confirmed by ^1^H-NMR spectra.

Figure 4 shows FT-IR spectra of poly(ε-caprolactone-co-D-lactide) star-shaped copolymer compared with neat PCL and PDLA to confirm star-shaped PCL-*b*-PDLA block copolymer. Compared to neat PCL and PCL-*b*-PDLA, spectra of neat PDLA discriminatively showed absorption bands corresponding to C=O stretching mode of ester group at 1745 cm^−1^. Stretching vibration absorption of -C-O- was confirmed at 1080 and 1180 cm^−1^. Symmetric bending of corresponding to C-H group and the symmetric bending of -CH_3_ were confirmed at 1450 cm^−1^ and 1360 cm^−1^, respectively (Figure 4a) [57].

In Figure 4a, both neat PCL and PCL-*b*-PDLA identically showed stretching vibration of C=O at 1720 cm^−1^. The asymmetric stretching peak and the symmetric stretching peak of CH_2_ corresponded to peaks at 2940 cm^−1^ and 2865 cm^−1^, respectively. Compared to neat PCL, PCL-*b*-PDLA copolymers showed main absorption bands corresponding to OC=O and C-OH at 1755 cm^−1^ and 1090 cm^−1^ (Figure 4b,c) [58].

#### 3.2.1. Mechanical Properties

Many researchers have made extensive efforts to modify mechanical performances of PLA and PLGA using plasticizers, blends, and composites. The present method blended PLGA with biodegradable plasticizer to improve the toughness of PLGA with considerable increase of elongation. Testing was conducted for PLGA films and PLGA/star-shaped PCL-*b*-PDLA plasticizer blended films. In this study, different molecular weight types of star-shaped poly (ε-caprolactone-co-D-lactide) plasticizers were synthesized and blended with PLGA. Three different types of plasticizers were blended with PLGA with the solvent casting method at room temperature.

The Figure 5 showed X-ray diffraction curves of neat PLGA, PLGA/star-shaped PCL-*b*-PDLA, PLGA/star-shaped PCL-*b*-PLLA, and synthesized PCL-*b*-PDLA plasticizer. The XRD pattern of PLGA/PCL-*b*-PLLA exhibited diffraction peak at 16.5°, 19.4° and 22.5°. Compared to PLGA/PCL-*b*-PLLA, the PLGA/PCL-*b*-PDLA showed extra peaks at 12°, 20.5°, and 24.1°, showing the presence of stereocomplexation (SC) between PLLA and PDLA [38,59]. The XRD pattern of star-shaped PCL-*b*-PDLA plasticizer showed the characteristic diffraction PCL reflections as appearing at 21.7° and 23.9° [60,61].

Figure 6a shows comparison of mechanical properties between neat PLLA PDLA and PLGA copolymers. Films were casted by solvents. As shown in Figure 6a, PLGA exhibited higher tensile strength, tensile modulus, and high elongation than PLLA and PDLA attributed to the inherently superior mechanical properties of PGA. With only 30% of glycolide in the copolymer, tensile strength of PLGA was up to 58 MPa, whereas neat PLLA and PDLA had tensile strengths up to 47–50 MPa.

Figure 6b shows mechanical properties in the form of stress-strain plot of PLGA with addition of poly(ε-caprolactone-co-D-lactide) Mw 5000 plasticizers. There were no significant transitions of modulus values between neat PLGA and PLGA blended with PCL-*b*-PDLA plasticizer. PLGA blended with 0.25% showed a slight increase of elongation from 52% to 88%. As shown in Figure 6b, when the addition amount of plasticizer was 0.5% to 1%, the result indicated a mechanical transition of the PLGA/PCL-*b*-PDLA plasticizer from brittle to ductile with remarkable improvement in its toughness. Addition of 0.5% and 1% PCL-*b*-PDLA to PLGA increased the strength yield by up to 248% and 281%, respectively. The PLGA/PCL-*b*-PDLA 0.5% plasticizer showed tremendous improvement of toughness, as 248% elongation was achieved without considerable loss in modulus or tensile strength. This could be due to a strong adhesion between PLGA main matrix and PDLA segment in PCL-*b*-PDLA composite. This improvement of PLGA was contributed to immiscibility and the presence of a higher density of intracrystalline connection between PLLA in PLGA and PDLA in PCL-*b*-PDLA. Tie et al. have investigated the effect of multi-armed polyester-*b*-PCL and emphasized that branched PCL polymer can improve mechanical properties of the main matrix [62].

However, mechanical properties of PLGA deteriorated when the addition of 5% PCL-*b*-PDLA as shown Figure 6c. The deteriorated phenomenon was due to the occurrence of chain scission in cross-linked chain from PLGA with PCL-*b*-PDLA plasticizer [63]. The result showed that the strain increased without a change in stress when the amount of biodegradable plasticizer was added up to 1%. Mechanical properties dramatically aggregated when the addition amount of plasticizer was increased to 5% as shown in Figure 6b. As shown in Figure 6c, similar behavior was observed with the addition of star-shaped PCL-*b*-PDLA Mw 10,000 to PLGA. Compared to Figure 6b, the value of elongation at break was increased. As shown in Figure 6c, PLGA/PCL-*b*-PDLA 0.5% had a stress value of 48 MPa and 271% of elongation at break. However, as the amount of added PCL-*b*-PDLA plasticizer increased, the strength value was sharply decreased, although it exhibited a very high elongation at break. These results demonstrate that increasing PCL-*b*-PDLA content can reduce the overall crystallinity of PLGA due to increase of amorphous portion in PLGA blended with PCL-*b*-PDLA plasticizer.

Figure 6c shows fractured sections along the tensile direction for PLGA and star-shaped PCL-*b*-PDLA Mn 15,000. The result showed enhancement of elongation at break but a decrease of stress. The deterioration of stress was due to aggregation of the high molecular weight of PCL-*b*-PDLA within the PLGA matrix to form larger particles. This result indicates that high molecular weight of spherical composites behaves as a stress concentrator and debonding in the initial stage of stretching at the particle-matrix interface.

#### 3.2.2. Thermal Properties

Figure 7a shows thermal transitional properties of PLGA and PLGA blended with star-shaped PCL-*b*-PDLA films by DSC analysis. The neat PLLA of melting crystallizations transition was confirmed at around 170 °C. Glass transition and crystallization transition were observed at 49 °C and 170 °C respectively. Compared to neat PLLA, PLGA showed glass transition at around 60 °C. With addition of star-shaped PCL-*b*-PDLA composites, the glass transition temperature of PLGA blended with star-shaped PCL composites changed as shown in Figure 7a. From Figure 7a, it was verified that the grass transition temperature was lower in accordance with addition of PCL composites.

As shown in Figure 7b, thermal stability of PLGA and PLGA blended with biodegradable star-shaped PCL composites was confirmed by TGA analysis. TGA trace graph (Figure 7b) showed that the thermal stability of the PLGA blended with star-shaped PCL-*b*-PDLA was slightly enhanced compared to that of neat PLGA. The thermal degradation behavior of the PLGA blended with star-shaped PCL-*b*-PDLA was changed due to effect of stereocomplexation between PLLA segment and PDLA segment in PLGA and PCL-*b*-PDLA matrix. This result means that the structure is changed in PLGA matrix in accordance with addition of star-shaped PCL-*b*-PDLA composites. The stereocomplexation barrier in PLGA matrix can prevent heat and mass transportation during decomposition in the main matrix.

As shown Figure 7b, it was also confirmed that thermal degradation of the neat PLGA started at 250 °C. Compared to neat PLGA, the weight loss of the PLGA blended with star-shaped PCL-*b*-PDLA started at 270 °C while that of star-shaped PCL-*b*-PDLA blended 2% and 5% films started at temperatures greater than 280 °C. Figure 7b confirms that the thermal stability of PLGA films is improved with addition of star-shaped PCL-*b*-PDLA composite.

#### 3.2.3. Morphology

Surface morphologies of PLGA and its blends are shown in Figure 8, Figure 9 and Figure 10. Clearly, neat PLGA (Figure 8a) showed a relatively smooth and flat fracture surface compared to Figure 8b. Compared to Figure 8a, the fracture surface for PLGA blended with star-shaped PCL-*b*-PDLA had a fibrillated morphology as shown in Figure 8b. As shown in Figure 8b, we can confirm the dispersed rubbery composite which displayed core-shell structure consist with PCL rubbery core and PDLA semicrystalline outer shell within PLGA matrix.

Moreover, Figure 8c confirmed that the stereocomplexation and strong interaction between PLLA in PLGA and PDLA were involved in the degree of toughness improvement and immiscibility between PCL rubber and PLGA matrix. The star-shaped PCL-*b*-PDLA composite can contribute to homogenous between two phased and enhanced toughness due to interact with the main polymer matrix as shown in Figure 8c [56]. Star-shaped PCL-*b*-PDLA could generate more physical entanglement with the PLGA matrix, thereby affecting the mechanical properties of the blends. Moreover, the formation of stereocomplexes increases the interfacial adhesion between rubbery-PLGA interface, which further contribute to the improved toughness.

Figure 9 shows cross-sectional morphology images of PLGA and PLGA blended with star-shaped PCL biodegradable plasticizers. By referring to Figure 8a, the neat PLGA without star-shaped PCL plasticizers showed less porosity and smooth surface compared to Figure 9b–d. However, after blending with star-shaped PCL-*b*-PDLA plasticizers, the porosity structure in PLGA matrix was increased compared to that of neat PLGA which had a smooth cross-sectional surface as shown in Figure 9b–d. Images shown in Figure 9b–d confirmed to the craze-formation via an increase of porosity in the PLGA matrix. The presence of craze can induce significant toughness improvement due to void formation in the matrix that is engaged in debonding at interface or cavitation inside rubber particles.

Figure 10a,b show cross-sectional images of tensile-fracture path of PLGA and PLGA blended with rubbery PCL composite. The cross-sectional of Figure 10b showed brittleness due to the brittle nature of PLLA in the PLGA matrix. Cross-sectional images of PLGA blended with plasticizer (Figure 10b) showed a ductility characteristic due to the grafted-rubbery gel in the matrix. Figure 10c,d show surface images of whitening area in tensile fracture. As shown in Figure 10c,d, the formed star-shaped PCL-*b*-PDLA gel was interconnected with PCL rubbery gel and the main matrix via grafting. Thus, the presence of grafted-rubbery gel can influence the degree of crystallinity in the main matrix.

#### 3.2.4. Transparency

Figure 11 shows UV-Vis spectra of PLGA and PLGA blended with star-shaped PCL-*b*-PDLA biocomposite. As shown in Table 2, the transmittance of PLGA was 79.52% in the center of visible light spectrum at 550 nm. The light transmission result showed no significance difference between PLGA and PLGA blended with 0.25% which was 75.78% at 550 nm. However, as the addition amount of star-shaped PCL-*b*-PDLA increased, transparency values decreased to 73.71%, 71.14%, and 68.21% at 550 nm, respectively. Results shown in Figure 11 confirmed that the neat PLGA was colorless and transparent compared to PLGA blended with PCL-*b*-PDLA composites which showed a slight transparency.

## 4. Conclusions

Biodegradable star-shaped PCL-*b*-PDLA plasticizers with different molecular weights were synthesized using natural originated xylitol as an initiator. Molecular structures were confirmed by ^1^H-NMR and FT-IR spectroscopy. Super-tough PLGA transparent films were prepared through a solution casting method. Effects of added star-shaped PCL-*b*-PDLA plasticizers on mechanical, morphological, and thermodynamic properties of PLGA/star-shaped PCL-*b*-PDLA blends were investigated. PLGA/star-shaped PCL-*b*-PDLA blends exhibited slightly enhanced thermal stability compared to neat PLGA. With only 0.5 wt% addition of star-shaped PCL-*b*-PDLA (Mn = 5000), the elongation at break of the PLGA blend reached approximately 248% without any considerable sacrifice over excellent mechanical strength or modulus of PLGA. Moreover, the morphology of PLGA blends confirmed stereocomplexation and strong cross-linked network between PLLA segment and PDLA segment, which effectively enhanced the interfacial adhesion between the star-shaped PCL-*b*-PDLA plasticizers and the PLGA matrix.

## Data Availability

Not applicable.

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
