# Peer review of "Super-Tough and Biodegradable Poly(lactide-co-glycolide) (PLGA) Transparent Thin Films Toughened by Star-Shaped PCL-b-PDLA Plasticizers"

_polymers, 2023, doi:10.3390/polym15122617_

Round 1
Reviewer 1 Report
In general, this manuscript does not present a sufficient scientific novelty. Many errors or inaccuracies are present in the text. (See my comments)
The manuscript only shows a compatibilization effect of a PLGA copolymer by PCL-b-PDLA oligomers. It puts forward the effect of a stereocomplexation between the PLLA and PDLA segments, which in my opinion is not proven, and probably non-existent.
For these reasons, I do not think that this manuscript can be accepted for publication.
Comments
-Some of the references in the introduction are not entirely relevant.
-P1: A large part of the figure on page 1 is not easily readable. Some chemical bonds are not visible.
The title mentions "Fully green Starshaped PCL-b-PDLA Plasticizers", PCL is not considered a "green" compound! caprolactone is prepared chemically.
-P2 l64: "However, PCL and PLA are known to be thermodynamically immiscible [31]". Reference 31 never mentions PCL! Moreover, immiscibility is dependent on molecular weights!
-P2 l64-66: Reference 50 uses crystalline PLLA and crystalline PCL of high molecular weight, which explains the immiscibility. In this manuscript the (random) PLGA is not crystalline at all and the PCL-b-PDLA has a low molecular weight. The two cases are therefore not comparable!
-P23 l82-83: the incorporation of PLLA (crystalline) into PLGA (amorphous) to increase the toughness is an evidence!
-Fig 2: the "a" NMR peaks of the PGA need to be amplified to better visualize them. Furthermore, it appears that there are many peaks and not one peak as indicated in the text! Where do these various peaks come from?
What do the values 0.05, 0.29 and 0.80 under the 3 NMR peaks mean? They cannot be integrations, it would be completely incoherent.
--A fundamental point not being discussed is to find out the proportions of PCL and PDLA and the average molar masses of each branch of the star. In the case of a MM of 5000, given that there are 5 branches, the MM of each branch is around 1000. Since it is stated that the proportion of PCL is much higher than that of PDLA, the PDLA chains have only few units (< 3-4). In addition, the proportion of PCL-b-PDLA is very low. Therefore, the probability of having PLLA/PDLA stereocopolymers is almost zero.
-P6 l205: “peak 3147 at range from ?? to ? 206 ? signal…”. Peak 3147??
-P6 222-224: “methine group protons of PDLA showed a doublet signal ? 222 ? at 5.2ppm”. It should be a quadruplet. On the other hand, the peak at 1.4ppm is a doublet...
-P6 L 225: “which proved that the PDLA block was connected to PCL”. This proves that PDLA chains are present but not related to PCL. There is always the possibility of homopolymer PDLA. Therefore, in my opinion, the 1H NMR does not confirm the formation of P(CL-co-D-lactide). To do this, a DOSY NMR should be performed. Similarly, FTIR does not confirm the formation of the star copolymer.
-Figure 5 shows nothing significant. It can be removed.
-P10 l341 and figure 7: DSC analysis shows no melting point corresponding to the formation of a stereocomplex (about 200°C)! The DSC of PCL alone and especially that of PCL-b-PDLA are needed to make relevant conclusions.
-P11-13 The figures for the morphology show a plasticizing effect, but do not prove the formation of PCL-b-PDLA structures or the effective formation of stereocomplexes.
No comment
Reviewer 2 Report
This manuscript describes on the synthesis, characterization and application studies of fully-green starshaped PCL-b-PDLA plasticizers. Based on the results (results are expected) described in this article, the reviewer cannot find significant novelty in this manuscript. Consider the reputation of this journal, this reviewer does not recommend this article to be published in this journal. Some suggestions are listed below.
(a) Some formats should be corrected. (missed punctuation, superscript, capital letter, wrong spelling, ….)
(b) In Scheme 1, did the initiator work like a linkage? The Scheme should be re-considered. Some statements should be added. In addition, the product is PLGA or PLLGA?
(c) In Scheme 2, catalyst should be added.
(d) In Table 1, the equivalent ratio between CL and LA should be added.
(e) In Fig 3, PCL-5000, 10000, 15000 should be corrected as PCL-b-PDLA-5000, 10000, 15000.
(f) The format of references. (no pages in ref 1)
This manuscript should be re-modeled if the authors intend to resubmit.
Reviewer 3 Report
This article reports the preparation of starshaped PCL-b-PDLA and its application as a plasticizer of PLGA to form super-tough PLGA material. As a result, it was demonstrated that the addition of PCL-b-PDLA to PLGA effectively improved the mechanical properties of PLGA. Those polymers should be useful for the development of biodegradable polymer materials. However, there are several points that should be revised, for example, the description of PCL-b-PDLA seems insufficient. I think that this article can be accepted for publication in Polymers, after revision of the following points.
1) In Figure 2, why the intensity of the signals b : c (0.80 : 0.05) is so different from 3 : 1? What means the numbers “188” and “189”?
2) For the synthesis of starshaped PCL-b-PDLA, the feed molar ratio of CL, xylitol, Sn(Oct)2, and DLA must be clearly described in the footnote of Table 1. The yield and composition (DLA content in the copolymer) of each PCL-b-PDLA must be included in Table 1. Also consider the effective digits.
3) The authors described “However, mechanical properties of PLGA deteriorated when the addition of PCL-b-PDLA to PLGA exceeded 5%. (line 302-303)” and “Mechanical properties dramatically aggregated when the addition amount of plasticizer was increased beyond 5% as shown in Fig. 6(b). (line 306-307)” However, no data for the PLGA/PCL-b-PDLA blend with PCL-b-PDLA content of higher than 5wt% are shown.
4) What is the “scissor-like reaction” (line 303-304)? It should be briefly explained with appropriate references.
5) The authors described “Mechanical properties dramatically aggregated when the addition amount of plasticizer was increased beyond 5% as shown in Fig. 6(b) (line 355-356).” The authors should provide some evidence to demonstrate the formation of stereocomplex in PLGA/PCL-b-PDLA blend.
6) In DSC analysis (Figure 7(a)), the melting transition of stereocomplex crystals was not clear for the PLGA/PCL-b-PDLA blends. I think that it should be mentioned in the main text.
7) The authors described “Moreover, Fig. 8(c) confirmed that the stereocomplexation and strong cross-linked network between PLLA in PLGA and PDLA were involved in the degree of toughness improvement and immiscibility between PCL rubber and PLGA matrix (line381-383).” From Fig.8(c), I understand that the addition of PCL-b-PLLA to PLGA resulted in the formation of some texture on the surface, but it is not clear whether the texture was made by stereocomplexation.
8) In this study, the authors applied the starshaped PCL-b-PDLA as a plasticizer of PGLA. I recommend the authors to discuss the effect of “starshape” of the copolymers by comparison with simple diblock or triblock copolymers. I also propose the authors to discuss the effect of “stereocomplex” by comparison with PCL-b-PLLA.
9) There are many careless mistakes in the manuscript. The authors should revise the manuscript carefully.
For example:
Line 199-200: “3” and “4” of “103” and “104” should be superscript.
Line 346: “From Fig. 6(a)” should be “From Fig. 7(a)”.
I recommend that the authors have English proofread.
Round 2
Reviewer 1 Report
The authors have answered many questions. However, in my opinion, one essential point remains unresolved: for me, there is no proof of the formation of a stereocomplex. It may exist, but its formation is never proven, it remains a hypothesis. this should be clearly stated in the text.
Furthermore, the authors' NMR correction on page 6, line 197 ff. is wrong: methine gives a quadruplet and methyl a doublet, not the other way round!
These cahanges must be made for the manuscript to be accepted
Author Response
"Please see the attachment"

Reviewer 2 Report
This reviewer recommends this manuscript to be published in this journal.
Author Response
Thank you for your comments.